# Metabolic Reprogramming of HCC: A New Microenvironment for Immune Responses

**DOI:** 10.3390/ijms24087463

**Published:** 2023-04-18

**Authors:** Beatrice Foglia, Marc Beltrà, Salvatore Sutti, Stefania Cannito

**Affiliations:** 1Unit of Experimental Medicine and Clinical Pathology, Department of Clinical and Biological Sciences, University of Torino, 10125 Torino, Italy; 2Institute for Research in Biomedicine (IRB Barcelona), The Barcelona Institute of Science and Technology (BIST), 08028 Barcelona, Spain; 3Departament de Bioquímica i Biomedicina Molecular, Facultat de Biologia, Universitat de Barcelona, 08028 Barcelona, Spain; 4Department of Health Sciences, Interdisciplinary Research Center for Autoimmune Diseases, University of East Piedmont, 28100 Novara, Italy

**Keywords:** HCC, metabolic reprogramming, immune response, glucose metabolism, fatty acid metabolism, amino acid metabolism, glutamine, urea cycle, TCA cycle, tumor microenvironment

## Abstract

Hepatocellular carcinoma is the most common primary liver cancer, ranking third among the leading causes of cancer-related mortality worldwide and whose incidence varies according to geographical area and ethnicity. Metabolic rewiring was recently introduced as an emerging hallmark able to affect tumor progression by modulating cancer cell behavior and immune responses. This review focuses on the recent studies examining HCC’s metabolic traits, with particular reference to the alterations of glucose, fatty acid and amino acid metabolism, the three major metabolic changes that have gained attention in the field of HCC. After delivering a panoramic picture of the peculiar immune landscape of HCC, this review will also discuss how the metabolic reprogramming of liver cancer cells can affect, directly or indirectly, the microenvironment and the function of the different immune cell populations, eventually favoring the tumor escape from immunosurveillance.

## 1. Introduction

Hepatocellular carcinoma (HCC) is the most common form of primary liver cancer, accounting for 80–90% of cases [1]. According to Global Cancer Incidence, Mortality and Prevalence (GLOBOCAN) 2020, HCC is the sixth most common cancer and the third most common cause of cancer-related mortality [2] worldwide with rapid growing global incidence (1 million individuals are expected to develop HCC by 2025 [3]) and a relative 5-year survival rate of approximately 18% [4]. Although the major risk factors for HCC remain hepatitis B virus (HBV) or hepatitis C virus (HCV) infections, the introduction of the HBV vaccine and the use of next-generation antiviral drugs should significantly reduce the progression of the disease to HCC [3]. By contrast, non-alcoholic fatty liver disease (NAFLD) is rapidly becoming a dominant cause of HCC as a consequence of the closed association between the pandemic, obesity and non-alcoholic steatohepatitis (NASH) and is predicted to amplify the incidence of HCC up to 41% by 2040 [2,5]. HCC denotes a poor clinical outcome often due to delayed diagnosis (with early stages being asymptomatic) when the majority of patients in advanced stages are then not eligible for curative therapy [2,6,7]. Liver transplant is the preferred treatment at the early stages, whereas surgical resection and trans-arterial chemo-embolization (TACE), as well as radiofrequency ablation (AFR), are the conventional Food and Drug Administration (FDA) approved approaches for advanced HCC. HCC shows high molecular heterogeneity and complexity due to different somatic mutations (in the TERT promoter, Tp53 and CTNNB1), the contribution of non-malignant cells of the tumor microenvironment (TME) (i.e., cells of innate and adaptive immunity [8]) and deregulated metabolism. As a result, HCC poses a challenge for global health care in identifying new therapeutic approaches and treatment strategies. Deregulation of metabolism is considered a common and relevant hallmark of cancers and heavily participates in promoting and supporting different events of tumor progression, including immune evasion [9,10]. Given the emerging relevance of metabolic rewiring in HCC and considering the liver cancer immune landscape, this review will describe the metabolic reprogramming of liver cancer cells, highlighting the newly discovered roles of some metabolites in modulating immune responses and cancer progression.

## 2. The Immune Landscape of HCC

HCC usually arises in chronically inflamed livers in which immune cells undergo functional reprogramming, resulting in dysfunctional immune responses that promote a cancer-prone microenvironment [11]. Such immune changes allow transformed cells to proliferate, leading to HCC development [12,13,14,15,16]. Therefore, it is easily conceivable that the amount, composition and differentiation stage of infiltrating immune cells impacts the HCC prognosis and its capability to respond to pharmacological treatments [1,17,18]. Because of its heterogeneity [19], the precise composition of the HCC immune milieu remains poorly characterized. However, the advent of cutting-edge techniques such as single-cell(sc)-RNA sequencing and mass cytometry paved the way for better defining the HCC immune microenvironment, providing novel insights concerning possible therapeutic targets [20,21,22]. Regardless of the etiology, the HCC immune landscape consists of multiple cellular components, including myeloid and lymphoid cells that interact with non-immunological cells such as cancer-associated-fibroblasts (CAFs) and liver sinusoidal endothelial cells (LSECs) among others, contributing to the loss of cancer immunosurveillance [1,23,24,25]. The HCC onset sees the recruitment and differentiation of innate and adaptive immune cells, displaying immunosuppressive or exhausted phenotypes and is therefore unsuitable in contrasting tumor growth [22,26,27]. The HCC microenvironment contains a heterogeneous population of immature myeloid-cell lineages, displaying variable degrees of differentiation, termed myeloid-derived suppressor cells (MDSCs). MDSCs have been classified as monocytic (M-) or granulocytic (G-) MDCSs since the former phenotypically resemble monocytes, while the latter resemble neutrophils [28]. Beyond the classification, MDSCs exert immunosuppressive functions on T cells by producing several mediators, including arginase 1 (ARG1), indoleamine 2,3-dioxygenase 1 (IDO1), transforming-growth factor β1 (TGF-β1) and interleukin (IL)-10 [29]. Aside from MDSCs, tumor-associated macrophages (TAMs) constitute another abundant cellular component of the HCC immune milieu [30]. TAMs represent an immunosuppressive subset of macrophages that produce a multitude of molecular mediators such as interleukin IL-10, TGF-β, programmed death-ligand 1 (PD-L1) and osteopontin (OPN), supporting HCC malignant progression [31,32,33,34,35,36]. Specifically, TAMs sustain angiogenesis, epithelial-mesenchymal transition (EMT) and cancer metastasis and inhibit T cell activation [30,37,38,39]. Furthermore, TAMs are involved in the recruitment of neutrophils that play a fundamental role in hepatic carcinogenesis [40,41]. Although an anti-tumoral action is generally ascribed to neutrophils, recent advances revealed the presence of tumor-associated neutrophils (TANs) displaying immunosuppressive properties [42]. TANs sustain HCC progression by expressing PD-L1 that, in its turn, binds the immune checkpoint PD-1 on T cells, transmitting inhibitory signals [43]. TANs also promote tumor metastasis by secreting a large amount of the pro-metastatic factor oncostatin M (OSM) [40,44]. Furthermore, TANs stimulate the differentiation and expansion of immunosuppressive Foxp3+ regulatory T cells (Tregs) that accumulate in the tumoral tissue, inhibit effector T cell functions and suppress anti-cancer immunity [45,46,47]. Along with this, the HCC immune microenvironment sees an enrichment of CD8^+^ and CD4^+^ T-lymphocytes showing an exhausted phenotype following the chronic antigen stimulation [27,48,49]. Furthermore, metabolic derangements occurring in HCC lower the fraction of CD4^+^ T-helper cells [50] and increase the amount of dysfunctional natural killer (NK) and NKT cells, favoring the loss of cancer immunosurveillance [51,52]. In parallel, plasma cell accumulation within the HCC microenvironment dismantles antitumor immunity by producing inhibitory mediators such as PD-L1 and IL-10 and inhibiting CD8^+^ T-cell activation [53]. Noteworthy is that the presence of infiltrating plasma cells increases cancer aggressiveness and lowers disease-free survival in human HCC [54,55].

## 3. A Comprehensive Picture of Liver Cancer Cell Metabolism

### 3.1. Alteration of Glucose Metabolism

#### 3.1.1. Glycolysis and Gluconeogenesis

Glycolysis is the major glucose (and glycogen) oxidation pathway, generating ATP, NADH and pyruvate as end products. Cancer cells strongly rely on glucose metabolism as their main energy source. Altered expression of glycolysis enzymes is a common characteristic of several types of tumor, including liver cancer [56]. The levels of some glycolytic enzymes increase in the cirrhotic precancerous liver and correlate with an elevated propensity to develop HCC [57]. The dependence of HCC on glucose has inspired the design of some anti-cancer treatments by targeting glycolysis. Therapeutic strategies intended to prevent the increase in HK2 expression or activity were efficient in preventing HCC growth in vitro and in vivo [58,59,60,61,62]. Similarly, pharmacological inhibition of phosphoglycerate kinase 1 (PGK1), a highly expressed enzyme in HCC cell lines, hinders proliferation by reducing aerobic lactate production [63].

The last step of glycolysis generates two molecules of pyruvate per glucose that can be catabolized through two different routes. During hypoxia or acute ATP demand (e.g., muscle contraction), pyruvate is quickly reduced by the lactate dehydrogenase (LDH), producing lactate and regenerating the NAD+ utilized in previous steps of glycolysis. Alternatively, under oxygen availability, pyruvate is incorporated into the tricarboxylic acid (TCA) cycle and is completely oxidized to CO_2_ in the mitochondria (discussed below). Although lactic fermentation is the less efficient option for ATP production, most cancer cells prime the utilization of glucose through this pathway even when oxygen is available (aerobic fermentation). This is a widely documented phenomenon in several types of cancers called the “Warburg effect” and allows cells to proliferate rapidly, favoring tumor formation and progression [64]. Initially considered a waste product of glycolysis, the lactate produced during fermentation has emerged as a contributor to tumor growth by inhibiting ferroptosis in Hep3B and Huh7 cells [65]. Lactate can also act as an epigenetic modifier by promoting histone lactylation [66]. Interestingly, a recent lactylome profiling on HBV-related HCC specimens reported high levels of lysine lactylation in enzymes of several metabolic pathways, including glycolysis, TCA, fatty acid metabolism, amino acid metabolism and drug metabolism [67]. Collectively, the available data indicate that high intracellular lactate levels have an effect on metabolic control during HCC progression.

Gluconeogenesis is the inverse anabolic pathway of glycolysis generating glucose from tricarboxylic molecules (pyruvate, lactate and glycerol) and glucogenic amino acids [68]. In HCC patients, gluconeogenesis is generally inhibited and the expression of some gluconeogenic enzymes decreases, correlating with poor prognosis. In particular, the levels of the rate-limiting enzyme phosphoenolpyruvate carboxykinase 1 (PCK1) are decreased in liver cancer. Additionally, cataplerosis by forced expression of PCK1 promotes the accumulation of ROS, leading to cancer cell apoptosis and hindering HCC progression [69]. Similarly, the expression of fructose 1,6-bisphosphatase 1 (FBP1) is lost in both human and murine liver cancer. Furthermore, the hepatocyte-specific deletion of FBP1 is sufficient to promote tumor progression [70].

#### 3.1.2. Pentose Phosphate Pathway

The pentose phosphate pathway (PPP) branches from glycolysis and provides reductive power (NADPH) and ribose. The PPP is a vital metabolic pathway during cell division as it provides building blocks for DNA synthesis and promotes redox homeostasis [71]. Consistent with the hyper-proliferative nature of cancer cells, increased protein levels of glucose 6-phosphate dehydrogenase (G6PD) and transketolase (TKT) were documented in HBV- and HCV-related HCC patients [72,73,74]. In particular, increased G6PD expression is associated with migration and invasion [72] and resistance to oxaliplatin chemotherapy [75]. 

#### 3.1.3. Tricarboxylic Acid Cycle

The TCA cycle is a series of biochemical reactions that oxidize acetyl-CoA into CO_2_, generating NADH and FADH_2_ that serve as electron donors to perform oxidative phosphorylation (OXPHOS) in the mitochondria. It represents a core of cellular metabolism where ATP, reductive potential and several macromolecule precursors are synthesized. Hepatic mitochondria are fundamental in many metabolic processes, including the clearance of toxic substances, gluconeogenesis and ketogenesis. Under oxygen availability, normal cells synthesize acetyl-CoA either from pyruvate via the pyruvate dehydrogenase (PDH) complex or by performing β-oxidation of fatty acids. The expression levels of the subunit PDHA1 are lower in HCC tumors compared with non-tumoral tissues [76], which is in agreement with the Warburg effect. However, the occurrence of the Warburg effect in cancer cells often generates the misconception that mitochondrial metabolism is dispensable for cancer cells. Accumulating evidence demonstrates that mitochondria are functional in cancer cells and that oxidative metabolism has a role during tumor growth [77]. While heavily lactylated, the protein levels of most TCA enzymes do not change in HCC when compared to adjacent liver tissue of HBV-related cohort [67]. Only the expression of succinate dehydrogenase (SDH) is frequently down-regulated in murine and human HCC, being associated with poor prognosis [67,78,79]. SDH inhibition leads to the accumulation of succinate and activation of the Yes-associated protein (YAP)/transcriptional coactivator with a PDZ-binding domain (TAZ) pathway, promoting HCC growth [79]. Supporting the idea that OXPHOS is important for cancer progression, inhibition of the ATP synthase (complex V) by using oligomycin synergizes with the anti-glycolysis drug 2-deoxyglucose (an analog of glucose) to promote breast cancer cells death [80]. Some HCC cell lines also show sensitivity to OXPHOS inhibitors in accordance with their basal mitochondrial respiration rates [81].

### 3.2. Alteration of Fatty Acid Metabolism

Fatty acids (FAs) are regarded as excellent storage compounds that can be used by cells as an energy source to grow and proliferate. Whereas normal cells rely on circulating exogenous lipids, HCC cells have a high degree of de novo lipid synthesis [82,83]. In fact, FAs operate not only as an energetic reservoir (provided through FAs β-oxidation –FAO-) but also as signaling molecules and structural components of the cell membrane, sustaining the high degree of proliferation of cancer cells, resistance to cell death and immunosuppression [84,85].

#### 3.2.1. Fatty Acid Synthesis

Diabetes mellitus, obesity and the metabolic syndrome are highly affected by the pathogenetic role of the de novo lipid synthesis. This, together with an increased intake of dietary FAs and enhanced lipolysis of visceral adipose tissue, determines a “lipid-rich condition” highly characteristic of obesity-driven liver cancer. Furthermore, HCC is characterized by a modulation of the Warburg effect and up-regulation of lipid metabolism. Therefore, enzymes and metabolites of FAs pathways have attracted growing attention in the perspective to elucidate mechanisms of tumor initiation and progression and to acknowledge new therapeutic strategies.

Many studies have reported that numerous enzymes involved in lipogenesis are up-regulated in HCC nodules, including the ATP citrate lyase (ACLY) [86,87], the acetyl-CoA carboxylase (ACC) [87,88], the fatty acid synthase (FASN) [86] and the stearoyl-CoA-desaturase 1 (SCD1) [89].

FASN, an enzyme that synthesizes a fundamental saturated FA (palmitate, C16:0) from acetyl-CoA and malonyl-CoA, is found up-regulated in numerous cancers. Its expression is usually associated with chemoresistance, metastasis and poor prognosis. FASN post-translational regulation through KAT8 acetylation-mediated proteasomal degradation was reduced in human HCC samples [90].

SCD is an enzyme able to generate monounsaturated fatty acids (MUFAs), mainly palmitoleate (C16:1) from palmitate and oleate (C18:1) from stearate (C18:0). It has been reported that SCD signaling is significantly associated with HCC progression and poor prognosis [82]. A recent study reported that oncoprotein hepatitis B X-interacting protein (HBXIP) enhances the expression of SCD, thus preventing sorafenib-induced ferroptosis and decreasing its anticancer activity [91].

The expression of all the lipogenic enzymes mentioned above is under the control of the transcription factor sterol regulatory element-binding protein (SREBP)-1. Large-scale gene expression profiling revealed significant up-regulation of SREBP1 in HCC samples and was associated with poor patient outcome [92]. 

#### 3.2.2. MUFAs and PUFAs

Phosphoglycerides (PGLs) are glycerol-based phospholipids that contain MUFAs or polyunsaturated fatty acids (PUFAs) as FA chains. They are the main component of biological membranes. Studies of the lipidomic profile in human samples of both NAFLD and HBV-associated HCCs reported a significant reduction of PGLs composed by PUFAs (such as arachidonic acid -C20:4-), often coupled by an increase in MUFAs containing ones (such as oleic acid -C18:1-) [93]. On the one hand, this lipid imbalance could be due to the high degree of PUFAs peroxidation associated with the typical HCC high oxidative damage burden. Consequently, low levels of PUFAs in membrane phospholipids could be a mechanism to avert ferroptosis [94]. On the other hand, higher levels of MUFAs are not only due to the overexpression of SCD1 [82,95,96] but also to the activity of fatty acid desaturase 2 (FADS2) that enables HCC cells to bypass dependence upon SCD1 for proliferation [97]. Interestingly, a recent lipidomic study showed that the unsaturated FAs significantly differ between NAFLD-HCC and other HCC patients with lower levels in NAFLD-HCC probably due to the increase of fatty acid transporters (CD36) that resulted in depletion of serum fatty acids [98].

#### 3.2.3. Cholesterol and Bile Acids

Cholesterol is a component of lipid membranes that regulates bilayer fluidity. It also plays an essential role in cellular signaling through the formation of cholesterol-rich membrane micro-domains named rafts. Cholesterol homeostasis is maintained by biosynthesis (mevalonate pathway), exogenous uptake, release and esterification [99]. Excessive dietary cholesterol intake was shown to play an important role in the development of steatohepatitis and is commonly referred to as an independent risk factor for the HCC development [100]. Furthermore, HCC exhibits a high number of rafts that can participate in the promotion of liver cancer cell proliferation and migration by up-regulating toll-like receptor 7 (TLR7) expression [101]. HCC is also characterized by an up-regulation of genes involved in cholesterol biosynthesis (HMG-CoA reductase –HMGCR-, mevalonate kinase –MVK-, squalene epoxidase –SQLE-) and catabolism (cholesterol 7 alpha-hydroxylase -CYP7A1-) correlating with poor prognosis [102,103,104]. In particular, SQLE was found to be overexpressed in human NAFLD-associated HCCs compared to normal controls [103] and its expression promotes HCC growth, EMT and metastasis by activating TGF-β/SMAD signaling [105]. A recent study showed that the tumor suppressor p53 participated in the stabilization of the sterol O-acyltransferase (SOAT1), an enzyme responsible for cholesterol ester biosynthesis that contributes to hepatocarcinogenesis by increasing cholesterol esterification [106]. This notion is further supported by a multiomic analysis conducted on HBV-positive HCCs, where the authors highlighted that SOAT1 was significantly enriched in HCCs of the most aggressive subclass, associated with the lowest overall rate of survival [104].

Bile acids (BAs) comprise a heterogeneous group of amphipathic steroid acids that function as dietary fat emulsifiers, favoring nutrient assimilation in the gut. BAs are classified as primary BAs when synthesized in the liver as a result of cholesterol catabolism, and secondary BAs when dehydroxylated by gut microbiota. Aside from their role in nutrient absorption, BAs can act as signaling molecules and regulate cellular metabolic, inflammatory and proliferative responses via a series of BAs receptors. Among them, farnesoid X receptor (FXR), Takeda G protein-coupled receptor 5 (TGR5) and G-protein coupled BA receptor 1 (GPBAR1) are widely involved in HCC development and progression [107,108,109,110,111,112]. Extensive literature data from in vitro and in vivo studies firmly indicate a pathological role for excessive BAs accumulation in the pathogenesis of HCC. Profiling of circulating BAs in the sera of HBV infection and NASH patients has revealed that increased levels of these molecules positively correlate with disease severity and a further increase in cirrhotic and/or HCC patients [113,114,115].

#### 3.2.4. Sphingolipids

Sphingolipids are a large and ubiquitous class of lipids, essential components of cell membranes and signaling molecules. They are synthesized in the endoplasmic reticulum by serine palmitoyl transferase (SPT), while further reactions lead to the formation of ceramides [116]. Ceramides are considered anti-cancer factors promoting apoptosis and cell cycle arrest [117,118] and are typically reduced in human HCC [119,120,121,122,123,124,125]. Furthermore, ceramides may be converted into more complex sphingolipids, such as sphingomyelin (SM), sphingosine 1-phosphate (S1P) and glucosylceramides (GlcCer) [93] which are considered pro-tumoral factors and whose production is enhanced during HCC pathogenesis [126,127,128,129,130,131,132]. Interestingly, a lipidomic analysis showed that SM levels are lower in HCCs from viral origin compared to HCCs from NAFLD origin [98].

### 3.3. Alteration of Amino Acid and Glutamine Metabolisms

Metabolic reprogramming in HCC also involves amino acids (AAs), fuel and engine for tumor growth; in fact, AAs represent building blocks not only for protein synthesis but also for nucleotides and other relevant biological molecules (i.e., heme) [10]. The liver is one of the central organs designed to control AAs metabolism, which appears strongly deregulated in HCC patients [133]. Moreover, AAs are deeply involved in modulating the immune response in the tumor microenvironment [134]. Therefore, their homeostasis acquires great relevance in the prevention and treatment of HCC, assuming a potential role as a biomarker and therapeutic target. In particular, HCC is usually accompanied by metabolic alterations in glutamine [133,135], asparagine [136], branched-chain amino acid (BCAA) [10,133,137] and urea cycle [10,138].

#### 3.3.1. Glutamine Metabolism

HCC undergoes a metabolic rewiring of glutamine homeostasis, acquiring the so-called “glutamine addiction” phenotype that is characterized by higher uptake of glutamine and consequent increased glutaminolysis [134,135]. Glutamine, the most significant non-essential amino acid for cells, is uptaken through the alanine-, serine- and cysteine-preferring transporter 2 (ASCT2, also called solute carrier family 1 member 5, SLC1A5) [139]. Glutamine catabolism occurs via a two-step process: a first conversion from glutamine to glutamate by the kidney-type and liver-type glutaminases (GLS1 and GLS2, respectively) followed by a deamination catalyzed by the glutamate dehydrogenase 1 (GLUD1 or GDH1), leading to α-ketoglutarate (α-KG) production, to fuel TCA cycle [139]. Normal hepatocytes preferentially express GLS2, whereas, during HCC development, a MYC-dependent metabolic switch from GLS2 to GLS1 occurs and sustains the rewiring of glutamine metabolism in tumor cells [139,140]. In particular, Yu et al. [140] showed a predominant and strong positive staining against GLS1 in HCC tumor cells (74.11%) in a cohort of 111 patients. By contrast, GLS2 is more abundant in hepatocytes of non-neoplastic surrounding tissue (92.7%) as compared to HCC nodules where most of the cases are negative (62.5%). Moreover, GLS1 and GLS2 are considered as independent prognostic factors for HCC, with GLS2 negatively correlated to tumor stage and overall survival, possibly for its anti-proliferative effects as a result of blocking G2/M phases of the cell cycle. On the other hand, high levels of GLS1 positively correlate with late-stage clinicopathological features, cancer stem cell phenotype and poor prognosis, potentially due to its ability to induce the pro-proliferative Akt/GSK3β/cyclinD1 as well as ROS/Wnt/β-catenin signaling pathways [135,138,140,141]. Concerning glutamine homeostasis, up-regulation of the specific transporter ASCT2 (SLC1A5) or the more generic L-type amino acid transporter 1 (LAT1, also known as SLC7A5), was observed in HCC tissue compared with adjacent non-tumor tissue, correlating with tumor size [135,138,139].

#### 3.3.2. Urea Cycle

The urea cycle, also called the ornithine cycle, is a process that takes place primarily in the liver and consists of the transformation of ammonia, a toxic final product derived from amino acid catabolism and/or protein degradation, into urea which is then excreted via urine [10,138]. This metabolic pathway is regulated by five essential enzymes: (i) carbamoyl phosphate synthetase 1 (CPS-1), the flow-producing urea cycle feed enzyme, whose role is to catalyze the conversion of ammonium into carbamoyl phosphate; (ii) argininosuccinate synthetase 1 (ASS1), that acts as rate-limiting enzyme generating argininosuccinate from citrulline; (iii) argininosuccinate lyase (ASL), the enzyme behind the conversion of argininosuccinate to arginine and fumarate; (iv) ARG1 which metabolizes arginine into ornithine and releases urea that can be finally excreted; (v) ornithine transcarbamylase (OTC) that localizes into mitochondria and converts ornithine and carbamoyl phosphate to citrulline [10,138,142,143]. In HCC patients, a significant reduction in genes of key enzymes of the urea cycle and associated metabolites (citrulline, arginine and ornithine) was observed. This peculiar signature was found to be clinically correlated with macrovascular invasion and poor prognosis [10]. In particular, CPS1 and ASS1 genes are epigenetically down-regulated in HCC through hyper-methylation events [10,138]. Inhibition of CPS1 by aflatoxin B1 results in a reduction of proliferation rate and increased apoptotic events in HCC cell lines [10]. Moreover, the silencing of ASS1 leads to smothering the endogenous arginine and turns HCC into an “arginine-auxotrophic” phenotype, making it necessary to increase the exogenous one uptake [10,144]. This metabolic vulnerability is being exploited as a prognostic biomarker and a therapeutic target. Consistently, the stable knockdown of ASS1 led to increased migration and invasion of HCC cells as well as enhanced HCC metastasis [138]. Along these lines, the over-expression of ARG1 affects liver cancer cell behavior by promoting typical morphological changes of EMT and increasing migration and invasiveness [138,145]. The last key enzyme of the urea cycle is the OCT, so much so that its loss of function or down-regulation leads to a gap in the urea cycle with consequent accumulation of ammonia in the blood [138,146]. Low expression of OTC correlates with poor overall survival and shorter disease-free survival in HCC patients [143]. Moreover, when focusing on phenotypic cell responses, in vitro experiments performed on Huh7 and SK-Hep-1 cell lines show that specific silencing of OCT with RNA interference promotes cell proliferation and colony formation [143].

#### 3.3.3. Branched-Chain Amino Acid

BCAAs are a family of essential amino acids that includes leucine, isoleucine and valine. These three amino acids, introduced through the diet or derived from protein, sustain cell proliferation, representing a nitrogen and/or carbon supply. The cytoplasmic branched-chain aminotransferase (BCAT1) and/or the mitochondrial branched-chain aminotransferase (BCAT2) isoenzymes transform BCAAs into corresponding branched-chain α-keto acids (BCKAs). This bioconversion consists in moving the α-amino group onto α-KG and producing glutamate, which in turn is a crucial metabolite for nucleotide and non-essential amino acids biosynthesis [147,148]. On the other hand, BCKAs are used as indirect sources of acetyl-CoA and succinyl-CoA that are then enrolled in the TCA cycle for ATP generation [147,149,150]. Concerning the liver, the role of BCAAs in HCC development and progression is a controversial topic that still needs further investigation. Available data reveal that BCCAs administrated for more than 6 months significantly decreases the incidence of HCC with a lower trend in liver-related events in patients with Child–Pugh A cirrhosis [151], showing beneficial effect in all stages of HCC [152]. Moreover, Kikuchi et al. reported that BCAAs supplementation reduced HCC recurrence in cirrhotic men with obesity (BMI > 25 Kg/m^2^) and serum levels of α-fetoprotein levels > 20 ng/mL [153]. These data are consistent with in vitro studies on HepG2 cells in which the addition of BCAAs to culture medium inhibited insulin-mediated proliferation of liver cancer cells by reducing the expression of insulin-like growth factor-1 receptor (IGF-1R) and increasing autophagy [154]. Other in vitro experiments have demonstrated that BCAAs administration in HepG2 cells cultured in high-insulin conditions leads to rapid VEGF degradation, through BCAAs-induced post-translational modification [155]. Takegoshi et al. reported that BCAAs supplementation in mice diet prevents HCC development by arresting the progression of non-alcoholic steatohepatitis [156]. By contrast, Ericksen et al. reported increased serum levels of BCAAs in HCC correlated with reduced BCAAs catabolism and enhanced mTORC1 pathway activation. Furthermore, the loss of BCAAs catabolism is then associated with HCC development, progression and lower overall survival [157]. 

#### 3.3.4. Asparagine Metabolism

Asparagine is a critical amino acid involved in glycoprotein synthesis that acts as a sugar-group binding site. The enzyme asparagine synthetase (ASNS) converts aspartic acid into asparagine [10]. ASNS expression is increased in HCC compared to the healthy liver, but its expression has decreased in parallel with the tumor stage, with low ASNS expression associated with poor prognosis [158]. Advanced HCCs are characterized by low levels of asparagine and are more sensitive to L-asparaginase (ASRGL1) therapy, promoting asparagine exhaustion in cancer cells [158]. ASRGL1 hydrolyzes L-asparagine to L-aspartic acid and ammonia. ASRGL1 expression was significantly increased in HCC tumor tissues compared to adjacent ones. Moreover, omics investigation performed by using the UALCAN database as well as analysis of the Kaplan–Meier plotter and Gene Expression Profiling Interactive Analysis (GEPIA) databases revealed a direct correlation between high ASRGL1 expression and poor clinical outcomes of HCC patients [159].

### 3.4. Nucleotide Metabolism

Nucleotide metabolism is critical for the biosynthesis of genetic material (DNA and RNA) and for cellular signalling, enzymatic regulation and metabolism. Proliferating cancer cells are characterized by increased de novo biosynthesis of nucleotides, requiring R5P generated from PPP and amino acids [10]. Available data indicate that HCC from mixed etiology (alcohol, HBV, HCV), shows up-regulation of genes involved in purine biosynthesis [160,161,162]. Moreover, up-regulation of the carbamoyl phosphate synthetase 2, aspartate transcarbamylase and dihydroorotase (CAD) gene, which catalyze the first three steps of pyrimidine biosynthesis, was observed in HCC [163,164]. Particularly, high fractions of HCC with CPS1 hypermethylation are characterized by increased activity of carbamoyl phosphate synthase II, which favors de novo pyrimidine synthesis from the shunting of glutamine [163]. In addition, an association between cancer stemness, poor prognosis of HCC and up-regulation of key enzymes of pyrimidine biosynthesis, such as deoxythymidylate kinase (DTYMK), thymidylate synthase (TYMS) and thymidine kinase 1 (TK1), has been described [165].

## 4. Metabolic Changes as a Result of HCC Etiology

A recent study by Boris et al. identified a unique cytokine blood profile in HCC patients depending on their aetiology, suggesting a different immune response depending on the HCC causative agents [166]. In addition to the understanding of metabolic alterations occurring in HCC, it is necessary to account for metabolic changes associated with specific risk factors that promote the insurgence of HCC. 

Concerning HBV infection, literature data indicate that HBV induces metabolic changes similar to those observed in HCC patients. In particular, HBV infection leads to dysregulation in lipid metabolism, presenting with the up-regulation of HBx, SREBP1, PPARγ and other adipogenic or lipogenic enzymes [102,167]. Consistent with those reported for HCC, HBV-related chronic liver diseases show an increase in triglycerides [168]. On the other hand, unlike the HCC, urea cycle metabolites such as citrulline and ornithine increase during the progression of chronic HBV [168]. In a similar way to HBV, HCV also mirrors some metabolic changes observed in HCC, supporting the Warburg effect by up-regulating glycolytic enzymes [169] and down-regulating oxidative phosphorylation [170,171]. Opposite to what is commonly observed in HCC, liver cancer cell lines infected with HCV virus result in up-regulation of gluconeogenesis [172]. Moreover, HCV leads to up-regulation of PPAR-γ, resulting in activation of fatty acid oxidation [169]. By contrast, there is little information on the metabolic alterations occurring in liver disease associated with alcohol consumption. However, similar to HCC, an increased expression of the fatty acid translocase CD36 seems an important trait involved in the development of alcoholic fatty liver disease [173]. Dysregulation of lipid metabolism is also the major distinctive hallmark of NAFLD, actually considered an emerging and predominant risk factor for HCC [174]. Beyond lipid metabolism, NAFLD patients show increased serum levels of BCAAs [175], potentially affecting glucose metabolism via mTOR signaling [176,177]. Although available data on alteration of BCAAs metabolism in HCC are, in some cases, contradictory, administration of BCAAs in NAFLD patients results in preventing HCC development [137]. Moreover, evidence-based guidelines suggest a dietary implementation with BCAAs for treating HCC [151,156]. Moreover, NAFLD also mirrors HCC metabolic rewiring in terms of mitochondrial oxidative metabolism, gluconeogenesis, lipolysis and anaplerotic pathways [178,179,180].

## 5. Effect of HCC Metabolic Reprogramming on Immunity

Tumor cells, including liver cancer cells, reprogram their metabolism to proliferate and disseminate [139]. These metabolic changes can affect the TME, including immune cells, often creating immune-suppressive conditions and contributing to tumor escape [181,182]. A summary of how metabolic rewiring of liver cancer cells impacts on immune cell responses is provided in this section.

Based on the current knowledge, changes in the levels of glycolytic and PPP enzymes are not associated with the accumulation of intermediates relevant to immunity. However, the accumulation of lactate as the main end product of glycolysis can affect tumor-associated immune responses. Lactate promotes the conversion of TAM from the anti-tumoral M1 phenotype to the pro-tumoral M2 one by activating the G protein-coupled receptor 132 (Gpr132) [183,184,185]. Similarly, TCA-derived metabolites act as signal transductors affecting immune cell function. Succinate, for instance, is sensed by macrophages, leading to increased IL-1β secretion by activating succinate receptor 1 (SUCNR1) and amplifying toll-like receptor (TLR) signaling [186]. Succinate is secreted to the medium by different cancer cell lines, promoting TAM differentiation and promoting cancer cell migration, invasion and metastasis [187]. In the liver, succinate is a relevant signaling molecule during liver fibrosis [188]. To our knowledge, no available data provide a direct connection between HCC progression and increased extracellular succinate levels. However, considering the increased intracellular succinate levels in HBV-related HCC [67], it is plausible that succinate has a role in TAM activation during liver cancer. Isocitrate dehydrogenase (IDH2) is a TCA enzyme that catalyzes the reduction of isocitrate to α-KG, generating NADH. Mutations in IDH isoforms can be found in some types of solid tumors, including liver cancer [189]. These mutations decrease the physiological enzymatic activity of IDHs (generating α-KG) and boost their capacity to transform α-KG into 2-hydroxyglutarate (2-HG). High levels of 2-HG alter cell epigenetics by inhibiting the demethylation activity of 2-oxoglutarate-dependent dioxygenases, accelerating oncogenesis [190]. Additionally, elevated levels of 2-HG are detected in the serum of patients with IDH1/2-mutated intrahepatic cholangiocarcinoma [191], suggesting that overproduction of 2-HG leaks to the TME. In the bone marrow, 2-HG activates NF-κB in stromal cells via a reactive oxygen species/extracellular signal-regulated kinase (ERK)-dependent pathway in bone marrow stromal cells [192]. Altogether, the current evidence implies that 2-HG might work as a potential regulator of the tumor-associated immune response in liver cancer. A summary of metabolic changes in glucose metabolism occurring in liver cancer cells and connections with immune cells are represented in Figure 1.

Lipid metabolism has been linked to the progression of different cancers; however, the mechanisms underlying the impact of lipid metabolism on the tumor immune response have yet to be fully understood.

In an interesting study, Ma et al. showed that the release of free linoleic acid (C18:2) by lipid-laden hepatocytes causes more mitochondrially-derived oxidative damage than other free FAs and mediates selective intrahepatic CD4^+^ T lymphocyte cell-death without any involvement of intrahepatic NK T cells or intrahepatic CD8^+^ T lymphocytes in NAFLD-related HCC [193].

Along with this, arachidonic acid seems to play a role in hepatic tumorigenesis by mediating the synthesis of prostaglandin E2 (PGE2), which was found significantly elevated in human HCC tissues [194,195]. In obesity-associated HCC mouse models, PGE2 production suppresses antitumor immunity through the prostaglandin E2 receptor 4 (EP4) on immune cells, dampening the production of pro-inflammatory mediators, such as IFNγ and TNFα, and stimulating the production of anti-inflammatory cytokines (e.g., IL10) and tumor-promoting cytokines (e.g., IL6) [195].

Conversely, the role of cholesterol in hepatocarcinogenesis still remains controversial. Qin et al. investigated the effects of hypercholesterolemia on HCC development in mice and found positive correlations between serum cholesterol levels and NK cell cytotoxic activity against cancer cells, suggesting that high serum cholesterol levels may suppress HCC tumorigenesis via NK cell activation [196]. However, a recent study shows that obesity-induced hepatic cholesterol accumulation leads to NKT cell dysfunction suppressing HCC immunosurveillance. In fact, the hepatic accumulation of cholesterol due to SREBP-2 leads to lipid peroxide accumulation in NKT cells and results in deficient cytotoxicity against liver cancer cells [52].

The oxysterol liver X receptors (LXRs) are nuclear receptors involved in the regulation of lipid and cholesterol homeostasis, activated in response to increased cholesterol levels. LXR activation has been demonstrated to counteract tumor progression in breast cancer, colorectal cancer and HCC where it represents a prognostic marker [197]. Nonetheless, the anti-tumor properties of LXRs activation remains controversial. Many studies have reported the effects of LXRs on cancers after short-term use of LXRs agonists or antagonists but only a few data are available on the chronic activation of LXRs on HCC. A recent study revealed that the chronic activation of LXRα isoform exhibits a pro-tumoral effect through the induction of monocytic myeloid-derived suppressor cells and down-regulation of cytotoxic T cells and dendritic cells in experimental HCC. In this model, the activation of innate immune suppression is the result of the LXRα-dependent up-regulation of IL-6/Janus kinase/STAT3 signaling and complement pathways, as well as the down-regulation of bile acid metabolism [198].

The gut microbiome is a major determinant of hepatic BA composition and numerous studies have shown that its dysregulation plays a key role in the pathogenesis of liver diseases and HCC. Of note, Ma et al. described a mechanism by which the gut microbiome uses BAs as messengers to control the chemokine-dependent recruitment of hepatic NKT cells. In fact, they showed that primary BAs induced the recruitment of NKT cells to the liver mediating anti-tumor immunity, while secondary BAs dampened NKT cells recruitment, favoring a pro-tumorigenic milieu [199].

The sirtuin (SIRT) enzymes catalyze various biochemical reactions to remove acyl groups from histone and non-histone substrates. Among them, SIRT5 is an important metabolic regulator that has been found down-regulated in human HCC samples [200]. Accordingly, a SIRT5 deficient murine model revealed that loss of SIRT5 determines hypersuccinylation and abnormally increased BAs biosynthesis in the peroxisomes of hepatocytes. This accumulation of BAs acts as a signaling mediator directing macrophage polarization towards an immunosuppressive M2 phenotype that favors tumor-initiating cells and HCC development [200]. According to current knowledge, changes in sphingolipid levels are not associated with relevant modulation of immunity in HCC. Lipid metabolism of liver cancer cells and how this metabolic reprogramming affects immunity are summarized in Figure 2.

Concerning the amino acids metabolism, Bai et al. [136] identified a subclass of HCC characterized by increased asparagine metabolism, which in turn correlated with worse prognosis. These changes concerned increased levels of ASCT2 and down-regulation of GLS2. Interestingly, this specific high-asparagine HCC subgroup exhibits an immune-suppressive TME, with marked levels of Treg, Th follicular cells, MΦ macrophage and memory B cell infiltration. By contrast, M1 macrophages, T γδ cells, resting memory CD4^+^ T cells, mast cell and naïve B cell infiltration was decreased. Moreover, the HCC subtype with active asparagine metabolism shows a higher Tumor Immune Dysfunction and Exclusion (TIDE) score due to the higher levels of immune checkpoint genes (including CTLA4, HAVCR2, LAG3, PDCD1, TIGHT), predicting weak immune-therapy actions. Still related to asparagine metabolism, ASRGL1 expression positively correlates with HCC tumor stages and promotes tumor development and progression by modulating immune infiltration. In particular, ASRGL1 transcript levels are strongly associated with the infiltration of different immune cells, including TAMs, macrophages, Treg cells, CD8^+^ T cells, B cells, monocytes, dendritic cells (DCs) and Th1 in HCC tissues [159]. However, these data need to be further investigated. In this connection, glutamine metabolism is essential for the proliferation of cancer cells and activation of CD8^+^ T cells, the critical lymphocytes with anti-tumor activity in the immune landscape of HCC. Chen et al. have described an imbalance of the glutamine nutrient partitioning between cancer cells and cytotoxic T lymphocytes. In particular, HCC patients showing a more active glutamine metabolism in cancer cells than CD8^+^ T cells were characterized by an immune-suppressive TME. This peculiar metabolic trait suggests that cytotoxic T lymphocyte activity can be impaired when the glutamine metabolism of liver cancer cells exceeds those of killing lymphocytes [201]. Moreover, the expression of SLC1A5 can be considered as a prognostic marker for HCC outcome in relation to the immune microenvironment. In fact, a positive correlation between SLC1A5 mRNA levels and immune infiltration was reported, involving macrophages, B cells, CD8^+^ and CD4^+^ T cells, neutrophils and DCs [201]. Consistently, expression of SLC1A5 in HCC positively correlates with Treg markers and M2 macrophage phenotypical markers. By contrast, SLC1A5 expression negatively correlates with markers of M1 macrophages [202]. Along these lines, Zhang et al. [44] performed a comprehensive analysis of HCC heterogeneity by integrated multiomic approaches and proposed a new immune-phenotypic classification of HCC. In particular, three subtypes of HCC with peculiar TME in relation to immune landscape, chemokines/cytokines and cellular metabolism, were identified. One of the identified subtypes is viewed as an immuno-competent subtype, considering the normal levels of infiltrating T cells found in the tumor despite low levels of B cells. This immuno-competent subtype of HCC is characterized by a hyperactive urea cycle, which provides more arginine supply for T cells survival and killing activity. However, their toxicity to cancer cells is still too low and would require additional enhancement [203]. A study by Zhao et al. [133], used Univariate Cox and LASSO regression analyses of amino acid metabolism genes extracted from the Molecular Signature Database to generate a predictive risk signature of HCC patients from the TCGA-LIHC and GSE14520 (GPL3921) datasets. The authors revealed that alteration in amino acid metabolism alteration in HCC patients affects the ratio of infiltrated B and T cells, influencing HCC prognosis. Moreover, HCC patients identified as “high-risk” were characterized by low levels of NK infiltrating cells, which in turn can recruit macrophages and Treg cells, favoring HCC progression. A portrait of deregulated metabolism of AAs and connection with immune cells infiltration is summarized in Figure 3.

In relation to the nucleotide alteration observed in HCC, literature data indicate that purine metabolism and biosynthesis can affect the immune microenvironment of HCC (Figure 4). In particular, by analyzing a cohort of 372 HCC on TCGA database and by using the CIBERSORT algorithm Yang et al. revealed that HCC patients with both high purine biosynthesis and metabolism (PBhiPMhi) are characterized by low levels of M1 macrophages and CD4^+^ T cells as well as high infiltration of helper T cells and M2 macrophages [204]. 

## 6. Conclusions and Future Perspectives

As an emerging hallmark of cancer, metabolic reprogramming in HCC grows in relevance to address new therapeutic strategies for HCC treatment. Systemic HCC therapies show several limitations for the intra-tumoral heterogeneity of HCC due to recurrent multi-lesion forms of HCC, each one with its own distinctive genomic and/or epigenetic as well as metabolic alterations. From the current literature, HCC shows several metabolic alterations, including increased glycolysis and consequent reduction of the TCA cycle, up-regulation of lipid metabolism and amino acid biosynthesis, to support liver cancer proliferation and progression. Of relevance, in contrast to cancer cells, immune cells do not undergo “Darwinian evolutionary pressure”, thus cannot acquire an intrinsic plasticity that allows them to adapt to the developing tumors. However, their differentiation, proliferation and activation are deeply dampened by the metabolic changes occurring in the HCC tumor microenvironment. As mentioned above, growing evidence (summarized in Table 1) shows how HCC cancer metabolic rewiring can favor the instauration of immune-suppressive conditions and contribute to tumor escape reaction from immunosurveillance. Many of these metabolic alterations and related immune responses have been investigated compartmentalized. However, a better overall picture could be achieved considering the cross talk and synergy among pathways and their relations with etiology and immune cytokine profile. On these bases, the actual challenge is to identify new therapeutic strategies for counteracting the immune evasion and restoring a “good” immunological hot tumor phenotype (T-cell inflamed).

## Figures and Tables

**Figure 1 ijms-24-07463-f001:**
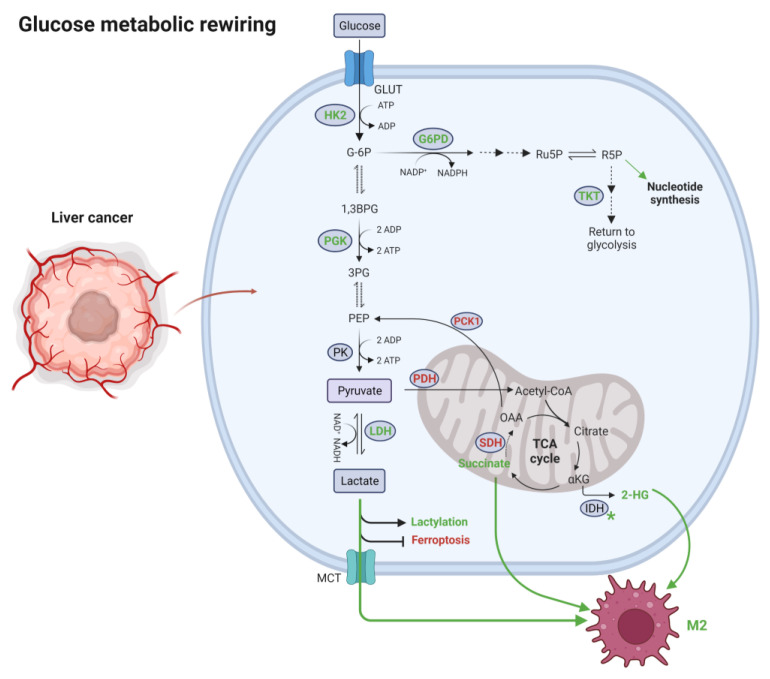
Metabolic rewiring of glucose in liver cancer cells and effects on immune cells infiltration. As illustrated in the review text, deregulation of glucose affects the recruitment and/or the activation of immune cells, potentially favoring immune escape of liver cancer cells. Abbreviations: 1,3BPG, 1,3-Bisphosphoglycerate; 2-HG, 2-hydroxyglutarate; 3PG, 3-Phosphoglycerate; G6P, glucose 6-phosphate; G6PD, glucose-6-phosphate dehydrogenase; GLUT, glucose transporter type; HK2, hexokinase 2; IDH, isocitrate dehydrogenase; LDH, lactate dehydrogenase; M2, M2 macrophage; MCT, monocarboxylate transporter; OAA, oxaloacetate; PCK1, phosphoenolpyruvate carboxykinase 1; PDH, pyruvate dehydrogenase; PEP, phosphoenolpyruvate; PK, pyruvate kinase; R5P, ribose 5-phosphate; Ru5P, ribulose 5-phosphate; SDH, succinate dehydrogenase; TKT, transketolase; αKG, α-ketoglutarate. Legend: green arrows and green bold text indicate up-regulated pathways; red arrows and red bold text indicate down-regulated pathways; * indicate mutations. Created with BioRender.com.

**Figure 2 ijms-24-07463-f002:**
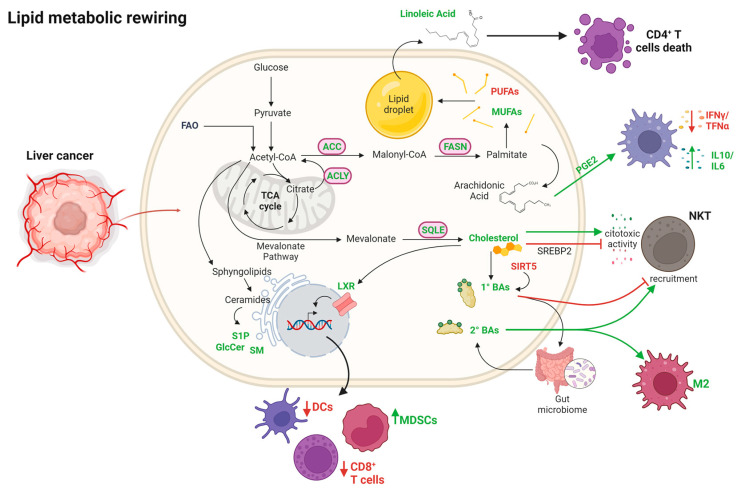
Metabolic rewiring of lipids in liver cancer cells and effects on immune cells infiltration. As illustrated in the review text, deregulation of lipids affects the recruitment and/or the activation of immune cells, potentially favoring immune escape of liver cancer cells. Abbreviations: ACC, acetyl-CoA carboxylase; ACLY, ATP citrate lyase; BAs: bile acids; DCs, dendritic cells; FAO, fatty acids oxidation; FASN, fatty acid synthase; GlcCer, glucosylceramides; M2, M2 macrophage; MDSCs, myeloid-derived suppressor cells; MUFAs, monounsaturated fatty acids; NKT, natural killer T cells; PGE2, prostaglandin E2; PUFAs, polyunsaturated fatty acids; S1P, sphingosine 1-phosphate; SM, sphingomyelin; SQLE, squalene epoxidase; SREBP2, sterol regulatory element-binding protein 2. Legend: green arrows and green bold text indicate up-regulated pathways; red arrows and red bold text indicate down-regulated pathways. Created with BioRender.com.

**Figure 3 ijms-24-07463-f003:**
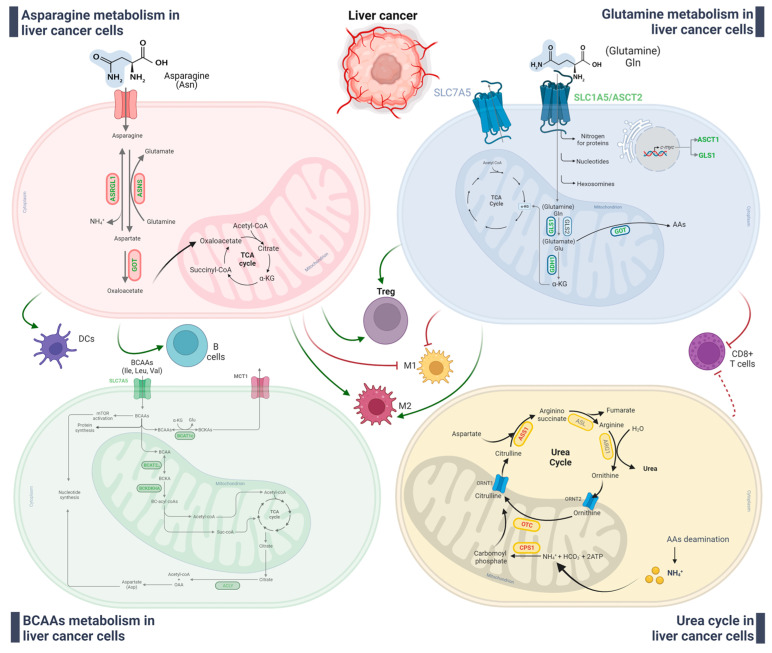
Metabolic rewiring of AAs in liver cancer cells and effects on immune cells infiltration. As illustrated in the review’s text, deregulation of AAs can affect the recruitment and/or the functionality of immune cells, potentially favor immune escape of liver cancer cells. Abbreviation: ASRGL1, L-asparaginase; ASNS, asparagine synthetase; GOT, aspartate aminotransferase; SLC7A5, solute carrier family 7 member 5; SLC1A5, solute carrier family 1 member 5 or ASCT2, alanine-, serine- and cysteine-preferring Transporter 2; GLS1/2, glutaminase 1/2; GDH1, glutamate dehydrogenase 1; α-KG, α-ketoglutarate; ARG1, arginase 1; ASL, argininosuccinate lyase; ASS1, argininosuccinate synthetase 1; OTC, ornithine transcarbamylase; CPS1, carbamoyl phosphate synthetase 1; ORNT1/2, ornithine transporter 1/2; BCAT1, branched-chain cytosolic aminotransferase; BCAT2, branched-chain mitochondrial aminotransferase; BCAAs, branched-chain amino acids; BCKAs, branched-chain α-keto acids; BCKDKHA, branched chain keto acid dehydrogenase; MCT1, monocarboxylate transporter 1; ACLY, ATP citrate lyase; DCs, dendritic cells; Treg, regulatory T lymphocytes. Legend: green arrows and green bold text indicate up-regulated pathways; red arrows and red bold text indicate down-regulated pathways. Created with BioRender.com.

**Figure 4 ijms-24-07463-f004:**
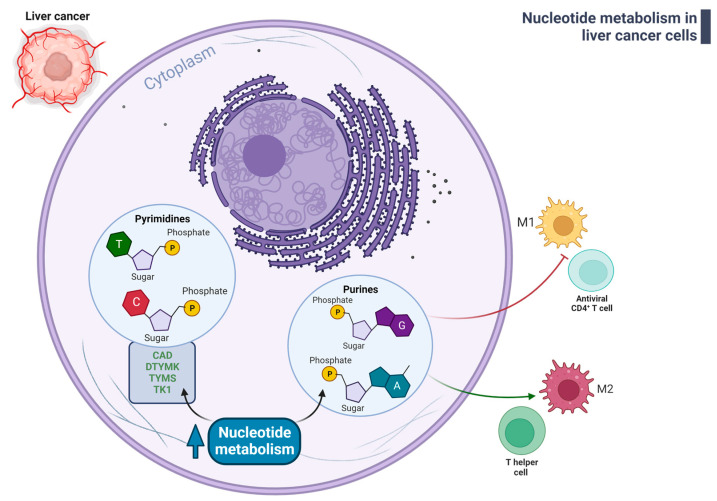
Deregulated nucleotide metabolism in liver cancer cells affects immune microenvironment. As illustrated in the review’s text, up-regulation of nucleotide metabolism is necessary to sustain liver cancer cells proliferation but can affect the recruitment and/or the functionality of immune cells, potentially favor cancer progression and/or immune escape of liver cancer cells. Abbreviation: CAD, carbamoyl phosphate synthetase 2, aspartate transcarbamylase and dihydroorotase (CAD); DTYMK, deoxythymidylate kinase, TYMS, thymidylate synthase, TK1, thymidine kinase 1. M1 and M2: M1 macrophages phenotype and M2 macrophages phenotype. Legend: green arrows and green bold text indicate up-regulated pathways; red arrows and red bold text indicate down-regulated pathways. Created with BioRender.com.

**Table 1 ijms-24-07463-t001:** Main relations between liver cancer cells rewiring and immune responses.

Metabolic Pathway	Metabolite	Receptor/Enzyme	Function	References
**Glycolysis**	Lactate		Favor M1 to M2 phenotypic switch	[183,184,185]
**Tricarboxylic acid cycle**	Succinate		Promote TAM differentiation and cancer cell migration, invasion and metastasis	[187]
	2-Hydroxyglutarate		Promote macrophages differentiation to M2 phenotype	[190,191,192]
**Lipid** **metabolism**	Linoleic acid		Mediate selective intrahepatic CD4^+^ T lymphocytes cell-death	[204,193]
	Prostaglandin E2		Reduce pro-inflammatory mediators (IFNγ and TNFα) and stimulate the production of anti-inflammatory cytokines (IL-10)	[195]
	Cholesterol		NK cell activation	[196]
	Cholesterol		Reduce cytotoxic activity against liver cancer cells	[52]
	Primary bile acids		Induce the recruitment of NKT cells	[200]
	Secondary bile acids		Reduce NKT cells recruitment favoring a pro-tumorigenic milieu	[200]
		↑ activity of oxysterol liver X receptor (LXRα)	Induce MDSCs and suppress of cytotoxic T cells and DCs	[198]
**Asparagine** **metabolism**		↑ ASCT2 ↓ GLS2	Create immune-suppressive TME: high levels of Treg, Th follicular cells, MΦ macrophage and memory B cells infiltration; low levels of M1 macrophages, T γδ cells, resting memory CD4^+^ T cells, mast cell and naïve B cells.	[136]
		↑ ASRGL1	Correlate with TAMs, macrophages, Treg cells, CD8^+^ T cells, B cells, monocytes, dendritic cells (DCs) and Th1 infiltration	[159]
**Glutamine** **metabolism**		↑ SLC1A5	Correlate positively with M2 and Tregs phenotypical markers and negatively with M1 macrophages markers	[201]
**Nucleotide** **metabolism**		↑ purine biosynthesis and metabolism	Correlate M1 macrophages and CD4^+^ T cells as well as high infiltration of helper T cells and M2 macrophages	[204]

## Data Availability

Not applicable.

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
