# Peer review of "Metabolic Reprogramming of HCC: A New Microenvironment for Immune Responses"

_ijms, 2023, doi:10.3390/ijms24087463_

Round 1

Reviewer 1 Report

This review presents a global landscape of how metabolic changes in HCC affect the tumor microenvironment for regulating immune cell responses. This paper is well written and organised in contents. I recommended it for the publication. Still, I also want to share my curiostiy about the link bewteen metabolic reprogramming of HCC and immune cell activation. What kinds of metabolites produced in HCC cells can activate or inhibit the surrounding immune cells? it will be better to provide a summary table for readers. Is it also possible for HCC to use its metabolites to hijack immune cells to join them to help defend against the exogenously delivered drugs? Sure, the specific metabolites are now still unknown. As known, cyclic dinucleotides are effective STING activators for immune therapy, it will be necessary to also discuss the content about the metabolism of nucleic acids in HCC.

Reviewer 2 Report

In the review "Metabolic Reprogramming of HCC: A New Microenvironment for Immune Responses", Foglia et al. provide information on the functional link between metabolic reprogramming in HCC and the tumour microenvironment. This well-written review addresses key points in the field of HCC with the aim of identifying new therapeutic targets against this cancer. This will certainly be of interest to people working not only in this particular field but also in other cancers.

However, I think there is an imbalance between the despcription of the different metabolic alterations and the link with immune cells. Part 3 gives a lot of information about the rewiring of metabolism that occurs in HCC, which is not always obviously linked to a change in the immune response. I belive that this review will gain in clarity if the authors focus on this fundamental aspect.

Different causative agents can induce HCC. Recently, a study identified a unique cytokine blood profile in HCC patients depending on their aetiology, suggesting a different immune response depending on the HCC causative agents (Cancers 2022, 14, 4900.https://doi.org/10.3390/cancers14194900). Although briefly mentioned in the introduction, the authors do not discuss the differences between these aetiologies in terms of their impact on metabolism. This could be potentially linked to a differential effect on tumor microenvironment and immune response.

Minor comments:

Lane 32: I think HCC is missing between ... 2020, and is.

Lane 264-265: Not only is repeated in the sentence

Round 2

Reviewer 2 Report

The authors properly addressed my comments and improved the manuscript accordingly. I really appreciated the Table1.

There is just a typo in lane 697: In particular or particularly but not In particularly.

Author Response

We would like to thank the reviewer for her/his overall positive evaluation of our manuscript. We have corrected spelling or syntax errors marked in blue.